# Illusory object recognition is either perceptual or cognitive in origin depending on decision confidence

**Josipa Alilović[1,2], Eline Lampers[1], Heleen A. Slagter[3,4], Simon van Gaal** [1,2]*

**1** Department of Psychology, University of Amsterdam, Amsterdam, the Netherlands, **2** Amsterdam Brain and Cognition, University of Amsterdam, Amsterdam, the Netherlands, **3** Department of Applied and Experimental Psychology, Vrije Universiteit Amsterdam, the Netherlands, **4** Institute for Brain and Behavior, Vrije Universiteit Amsterdam, the Netherlands

* s.vangaal@uva.nl

## Abstract

We occasionally misinterpret ambiguous sensory input or report a stimulus when none is presented. It is unknown whether such errors have a sensory origin and reflect true perceptual illusions, or whether they have a more cognitive origin (e.g., are due to guessing), or both. When participants performed an error-prone and challenging face/house discrimination task, multivariate electroencephalography (EEG) analyses revealed that during decision errors (e.g., mistaking a face for a house), sensory stages of visual information processing initially represent the presented stimulus category. Crucially however, when participants were confident in their erroneous decision, so when the illusion was strongest, this neural representation flipped later in time and reflected the incorrectly reported percept. This flip in neural pattern was absent for decisions that were made with low confidence. This work demonstrates that decision confidence arbitrates between perceptual decision errors, which reflect true illusions of perception, and cognitive decision errors, which do not.

## Introduction

Our perceptual experience can deviate substantially from the actual input that reaches our senses. Misperceiving or misinterpreting sensory input is a common characteristic of several neurological and psychiatric disorders, such as schizophrenia, but also during drug-induced hallucinatory states [1,2]. However, even in healthy people and under common circumstances, illusory perceptual experiences may occur, as illustrated by famous illusions such as the Müller-Lyer and the Kanizsa illusion [3]. Further, context and stimulus frequency can act as top-down priors that bias the interpretation of ambiguous visual information [4–6]. Yet, even in conditions without strong perceptual priors, perception can spontaneously differ from the veridical sensory input. In particular, when perceptual information is sparse or ambiguous, interpretation of the visual input is challenged, resulting occasionally in incorrect perceptual decisions [7,8]. The experiential aspect of such incorrect decisions is often inferred based on participants' behavioral report. However, even when one trust such introspective reports, the source of these decision errors remains unclear. Most importantly, with behavioral measures it

**Data Availability Statement:** The data and analysis scripts used in this article is available on Figshare: https://doi.org/10.21942/uva.c.6265233.v1.

**Funding:** This research was supported by a grant from the H2020 European Research Council (ERC

STG 715605 to SVG). The funders had no role in study design, data collection and analysis, decision to publish, or preparation of the manuscript.

**Competing interests:** The authors have declared that no competing interests exist.

**Abbreviations:** AUC, area under the curve; EOG, electrooculogram; FFA, fusiform face area; fMRI, functional magnetic resonance imaging; GAT, generalization across time; HC, high confidence; HEOG, horizontal EOG; LC, low confidence; LDA, linear discriminant analysis; MVPA, multivariate pattern analysis; OFA, occipital face area; ROC, receiver operating characteristic; SDT, signal detection theory; TOI, time-windows of interest; VEOG, vertical EOG.

is hard to disentangle whether decision errors originate from true illusions in perception (misperceptions) or stem from guessing behavior (misreports), although procedures to do so have made rapid progress [9–14]. Here, we investigated this issue and examined if misreported visual stimuli (e.g., face presented, house reported) were misrepresented at the sensory stage or the decision stage.

To determine the nature of misreports, we dissociate sensory from post-sensory or decision-related processes during human perceptual decision-making using electroencephalography (EEG) in combination with multivariate pattern analyses (MVPA). Forty participants performed a challenging perceptual discrimination task in which faces and houses were briefly presented, preceded, and followed by pattern masks, strongly reducing stimulus visibility. Participants discriminated which of the 2 object categories was presented and indicated how confident they were in this decision (high/low confidence) [15,16]. Discrimination correctness was titrated by individually adjusting stimulus duration to reach approximately 70% correct decisions, thereby inducing a large number of decision errors for our analyses (approximately 30%). Multivariate pattern classifiers were then used to characterize the time course of category-specific neural representations of correct and incorrect decisions, at different levels of confidence. Confidence measures served as a proxy for the vividness of perceptual experience [17], which allowed us to separate strong perceptual illusions (incorrect decision with high confidence) from weak perceptual illusions or mere guessing behavior (incorrect decision with low confidence) [7].

We experimentally dissociated perceptual from decision-related neural processes by employing 2 separate category localizer tasks: one uniquely tuned to sensory features of the images and the other sensitive to both sensory features and decision processes. Using a between-task MVPA generalization approach, we could track perceptual and decisional neural processes [18–20], while they evolved across time and how these processes differed for correct and incorrect perceptual decisions as a factor of confidence, and thus strength of perceptual experience. This strategy has a similar goal as recent approaches in the field of consciousness science, in which no-report paradigms have been introduced. In studies employing such no-report paradigms, neural processing is typically compared between experimental conditions in which participants are aware of specific stimuli, but do not have a task to do and thus merely passively perceive these stimuli, versus a condition where active report is required. Contrasting these conditions, usually also including an unaware condition is thought to separate neural signals associated with perceptual experience per se versus neural signals of post-perceptual processes (or task-relevance in the broader sense) [21–26]. We found that decision errors made with high confidence were associated with neural representations reflecting the misperceived object (e.g., a house could be decoded while a face was presented). This was the case even when the classifier was trained on a localizer task in which these objects were task-irrelevant and unattended, showing that such decision errors stem from perceptual illusions that are sensory in nature.

## Results

There were 2 experimental sessions. In the first, participants performed 2 category localizer tasks, while their brain activity was measured using EEG. The localizer tasks were used to train our pattern classifiers (LDA, linear discriminant analysis) [27]. In the sensory localizer task (**S1A Fig**), participants reported an infrequent contrast change of the central fixation dot (20% of trials). At the same time, streams of house and face images were shown at the center of the screen, which were fully task-irrelevant. In the decision localizer task (**S1B Fig**), a masked image of a face or of a house was presented and at the end of each trial participants indicated

which stimulus category they had perceived. The images were therefore task-relevant and attended. In the sensory localizer, the classifiers' sensitivity was thus tuned mainly to sensory features of the 2 stimulus categories due to the absence of attention/task-relevance of the face/house images [26,28,29]. Therefore, the sensory localizer is reminiscent of a so-called no-report paradigm [30,31], but then used to train localizers instead of being used as the main task of interest [22–24,32]. The decision localizer was, besides sensitive to sensory features, also sensitive to post-perceptual decision processes. Stimulus-response mappings were counterbalanced across blocks in the localizer tasks to prevent motor response preparation from systematically biasing stimulus category decoding. The orientation of the presented images was either left tilting or right tilting, with 50/50 likelihood. This feature of the stimulus was always task irrelevant and was used to test for differences in decoding between task-relevant (category) and task-irrelevant features (orientation) in the main discrimination task, as a function of decision correctness and confidence (see **Fig 1A** for an illustration of the main task).

In the second session of the experiment, EEG recordings were obtained during the main perceptual discrimination task, which was similar to the decision localizer task of the first session [33], but now images were presented shorter (20 or 30 ms) and performance was staircased during practice blocks in order to achieve approximately 65% to 70% discrimination accuracy. After each decision, participants also provided a confidence rating regarding their estimated accuracy of their face/house discrimination response. Discrimination performance was kept low to elicit many incorrect perceptual decisions and hence to induce enough misreports for subsequent analyses.

Participants correctly discriminated faces and houses in 69.47% of all images (SD = 7.48) in the perceptual discrimination task. There was a slight tendency to respond house more often than face (58% versus 42%), similarly to our previous study [33]. As expected, signal detection

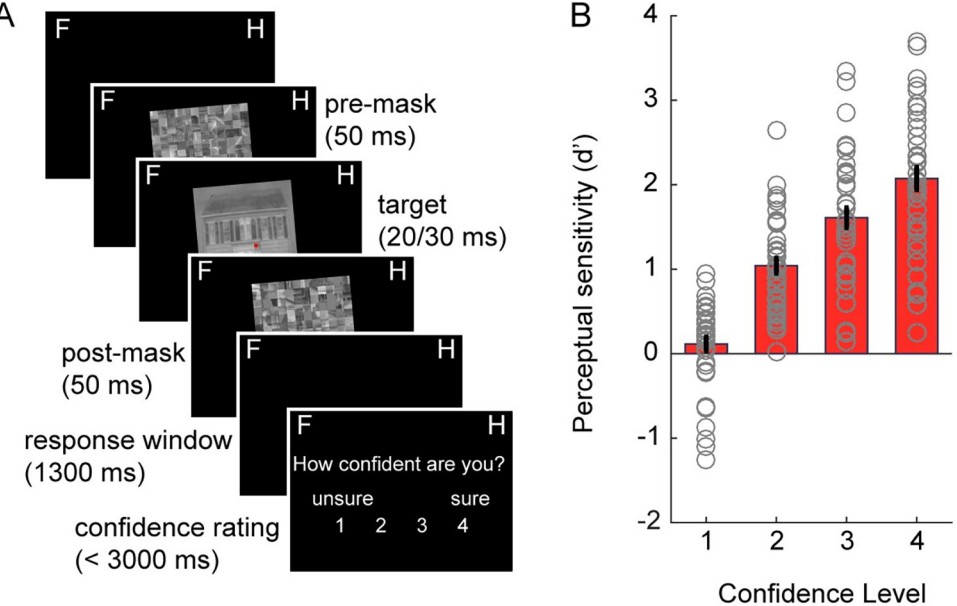

**Fig 1. Task and behavior.** (**A**) Each trial started with a central red fixation dot after which a forward mask was shown, followed by an image of a face or a house and a backward mask. Images were either tilted to the right or to the left (task-irrelevant feature, 5˚ or 355˚ angle). Note that in the example trial, only a left-tilted house image is shown. Participants reported whether they perceived a house or a face and indicated their confidence in this decision. (**B**) Perceptual sensitivity (d') as a function of decision confidence. The underlying data and scripts supporting this figure can be found on FigShare (https://doi.org/10.21942/uva.c.6265233.v1).

theory (SDT)-based perceptual sensitivity (d') increased with reported confidence level (main effect of confidence: $F_{3,117} = 116.69$, $p < 0.001$, $\eta^2 = 0.54$, **Fig 1B**) and ranged from d' = 0.11 at confidence level 1 (least confidence) to a d' of 2.07 at confidence level 4 (most confidence).

## The time course of category representations: Decision localizer

To investigate the time course of category-specific neural representations, we trained classifiers on EEG data recorded during the 2 localizer tasks and applied them to the main perceptual discrimination task data. We will first report the classification when classifiers were trained on the decision localizer (**Fig 2A**) and then the cross-classification to the main task (**Fig 2B**). Category-specific neural representations could be decoded based on the decision localizer (10-fold validation scheme, i.e., within-task decoding, **Fig 2A**). The generalization across time (GAT) matrix, time-locked to stimulus onset, exhibited the expected mixture of transient and stable on-diagonal decoding as well as persistent off-diagonal decoding profiles observed previously [33–37]. Following the analysis approach by Weaver and colleagues [33], who used a similar face/house discrimination task, we used matching time-windows for statistical analysis as used in that earlier study. For these analyses, based on Weaver and colleagues [33], decoding was examined on several stages, first of which a transient peak in an early time-window between 150 and 200 ms (**Fig 2A**, inset 2) [11,13–15]. The timing (150 to 200 ms; the decoding peak was found at approximately 166 ms) and scalp topography of the early peak is probably related to the N170 ERP component, often related to face processing [36,38–40], although also observed for letters, words, and biological motion [41,42]. Peaks of N170-like decoding have been associated with neural processes in the occipital face area, superior temporal sulcus, and/ or the fusiform face area (FFA) in ventral-temporal cortex [43–49]. Previous work has also shown that the strength of this early face-selective component is proportional to the strength (e.g., phase coherence of the stimulus) of the presented stimulus [18,40].

We also focused on a more stable square-shaped decoding profile between 350 and 500 ms (on-diagonal), capturing late category-specific processes (**Fig 2A**, inset 3), with a central-parietal

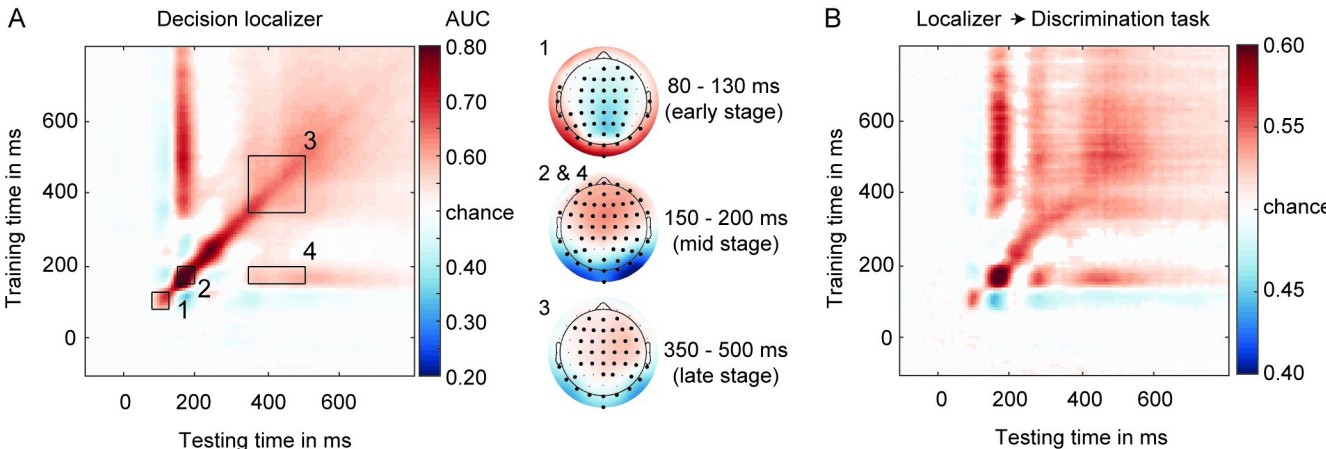

**Fig 2.** (**A**) GAT matrix for the decision localizer (all electrodes) and ROI marked by inset black boxes (numbers 1–4). On the right, covariance/class separability maps for each diagonal ROI, indicating the underlying data contributing to the classification performance. Highlighted channels significantly contributed to the classification performance after cluster-based permutation testing over channels (corrected for multiple comparisons). Covariance/class separability maps 2 and 4 are the same because decoding scores within insets 2 and 4 were obtained using the same training time-window. (**B**) Classifiers trained on the decision localizer applied to the main discrimination task (cross-task validation procedure, all electrodes), revealing a high degree of generalizability between the 2 tasks. The underlying data and scripts supporting this figure can be found on FigShare (https://doi.org/10.21942/uva.c.6265233. v1). AUC, area under the curve; GAT, generalization across time; ROI, regions of interest.

topography (late-stage topography). This decoding profile has previously been suggested to reflect global ignition, large-scale feedback processes allowing information to be broadcasted throughout the entire brain, making information explicit for report and decision-making [8,34,37,50].

Additionally, we examined decoding accuracy in a late time-window from 350 to 500 ms, based on training on early 150 to 200 ms classifiers (i.e., off-diagonal decoding, (**Fig 2A**, inset 4)). Given that decoding is in this case based on early classifiers tuned to sensory features, this stage is thought to reflect longer lasting sensory processes of categorical information across time [33,34,51]. This late-latency off-diagonal decoding pattern, referred to as the "perceptual maintenance" stage, has been suggested to reflect reactivation of early sensory stages through feedback processes [33–35].

Finally, although decoding was pronounced during 3 time-windows observed previously by Weaver and colleagues [33] (insets 2, 3, and 4), it was also observed at an even earlier time-window (80 to 130 ms) with a prominent occipital–parietal topography (**Fig 2A**, inset 1, early-stage topography). Because this early effect was absent in the same task when images where presented vertically [33], this early peak likely reflects differences in the orientations of the images presented, which we varied systematically across trials (note that training and testing was performed on balanced set of left and right oriented images). Although we follow a confirmatory approach by selecting time-windows of interest (TOI) beforehand [33], we will always in addition use across time permutation tests with multiple comparisons corrections to test for (unexpected) effects observed at other time-windows and to further specify the approximate onset and/or duration of our significant effects (see Materials and methods for details).

Cross validation between the decision localizer and the main discrimination task (**Fig 2B** for all channels; see **S2 Fig** for occipital–parietal channels) revealed a highly similar decoding profile for the 2 tasks. Classifiers' performance in differentiating between face and house images were highest on occipital–parietal electrodes, particularly during early latencies (**S2 Fig**). Since our aim was to examine whether decision errors have a perceptual origin, we report the decoding results based on data from occipital–parietal channels here to maximize sensitivity to neural signals that distinguish between the 2 categories. In the S1 Text, S4 Fig, and S5 Fig, we also report the critical decoding analyses using all channels, which yielded similar results.

## Diagonal decoding based on decision localizer

Next, we examined decoding scores as a function of decision correctness (correct/incorrect) and confidence (low confidence (LC): level 1 and 2 versus high confidence (HC): level 3 and 4, as in [33]). This created 4 conditions: HC correct, LC correct, HC incorrect, and LC incorrect trials. GAT matrices and on/off-diagonal time courses for these 4 conditions are shown in **Fig 3A–3D**. Note that the "testing labels" for classification were based on the decision of the subject in the main task (deciding face or house), not the presented stimulus. For correct trials, the presented stimulus and the decision are by definition the same. For incorrect trials however, the stimulus and the decision are of different categories. This means that if decoding scores are observed that are higher than chance level, the underlying data pattern correlates with the category of the reported stimulus, the decision. We will refer to this situation as positive decoding (with respect to chance level). In contrast, when below chance level decoding is observed, this would reveal that the underlying data pattern correlates with the presented stimulus, not with the reported stimulus (stimulus and response are different on incorrect trials, see Methods for details). The classifier thus finds negative evidence for the decision, and therefore we will refer to this situation as negative decoding (with respect to chance level).

Statistical analyses were performed on the diagonal and off-diagonal time courses (late sensory traces) (**Fig 3C and 3D**). We first report the analyses for the diagonal time courses using a 3-way

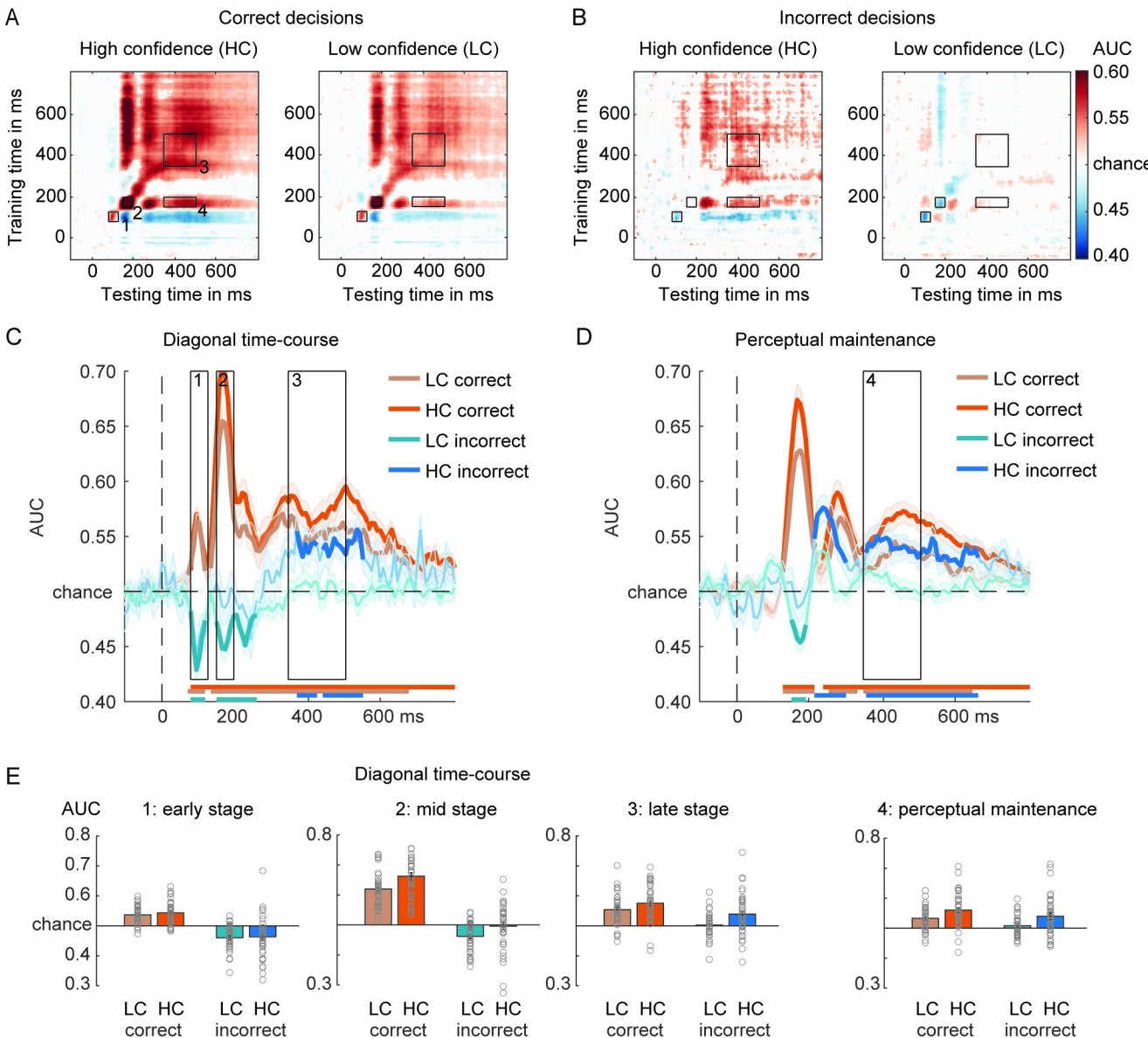

**Fig 3. Face/house classification in the main task while training on the decision localizer.** (**A**) GAT matrices for correct trials and (**B**) incorrect trials in the main discrimination task (occipital–parietal channels). Trials are separated by confidence—HC and LC—in the decision. (**C**) On-diagonal and (**D**) off-diagonal (perceptual maintenance) time courses of correct and incorrect decisions, separated by confidence. Black boxes indicate the time-windows (early, mid, late) used for averaging the AUC scores. Colored horizontal lines indicate periods of significant decoding with respect to chance. Longer-lasting sensory traces are derived by training a classifier on the time-window 150–200 ms and testing it across the entire time-window (panel D). Classification performance was evaluated at each time point using cluster-based permutation testing (two-tailed cluster-permutation, alpha $p < 0.05$, cluster alpha $p < 0.05$, N permutations = 1,000). (**E**) Bar plots show average AUC values for the TOI highlighted in panels (**C**) and (**D**). The underlying data and scripts supporting this figure can be found on FigShare (https://doi.org/10.21942/uva.c.6265233.v1). AUC, area under the curve; GAT, generalization across time; HC, high confidence; LC, low confidence; TOI, time-windows of interest.

repeated measures ANOVA with the factors Latency, Correctness, and Confidence. As anticipated, category decoding was better for correct than incorrect trials ($F_{1,39} = 135.05$, $p < 0.001$, $\eta_p^2 = 0.78$) as well as for HC versus LC trials ($F_{1,39} = 27.75$, $p < 0.001$, $\eta_p^2 = 0.42$). Category decoding differed across the 3 (early, mid, late) diagonal decoding time-windows ($F_{2,78} = 32.16$, $p < 0.001$, $\eta_p^2 = 0.45$), with differences being modulated by the level of confidence (Latency × Confidence:

$F_{2,78} = 6.88$, $p = 0.002$, $\eta_p^2 = 0.15$; no evidence for an interaction between confidence and correctness: $F_{2,78} = 0.022$, $p = 0.88$, $\eta_p^2 = 0.001$). As hypothesized, the characteristic pattern of decoding across 3 diagonal time-windows differed between correct and incorrect trials (Correctness × Latency interaction: $F_{2,78} = 36.57$, $p < 0.001$, $\eta_p^2 = 0.48$), independent of confidence (Correctness × Latency × Confidence interaction: $F_{2,78} = 1.12$, $p = 0.33$, $\eta_p^2 = 0.03$). As shown in **Fig 3C**, diagonal decoding profiles were very different for correct and incorrect decisions, especially for the early and mid-processing stages. Most prominently, it can be observed that for correct trials, all processing stages show positive decoding, whereas in sharp contrast the 2 early stages for the incorrect trials showed negative decoding. This negative decoding pattern on incorrect decisions illustrates that early EEG data patterns look more similar to the presented stimulus category than to the reported stimulus category (note again that the decisions were used as the classifier labels). The fact that the late stage decoding flips to positive decoding illustrates that later EEG data patterns are more similar to the reported category than the presented category (**Fig 3C**). We will come back to these effects later in the Results section.

For correct trials, decoding strength differed across the 3 TOI's ($F_{1,39} = 47.01$, $p < 0.001$, $\eta^2 = 0.55$) and was overall better for HC trials than for LC trials ($F_{1,39} = 31.91$, $p < 0.001$, $\eta^2 = 0.45$). Confidence interacted with latency (TOI's) ($F_{2,78} = 10.28$, $p < 0.001$, $\eta^2 = 0.21$), indicated by better decoding in the mid/late processing stage for HC versus LC trials (mid-stage: $t_{39} = -6.92$, $p < 0.001$, $d = -1.09$; late stage: $t_{39} = -3.06$, $p = 0.004$, $d = -0.48$; **Fig 3C**), while we did not find evidence for this confidence modulation for the early processing stage ($t_{39} = -1.25$, $p = 0.22$, $d = -0.2$). These results are summarized in the bar plots of **Fig 3E**.

On incorrect decisions, decoding scores across 3 diagonal time-windows differed ($F_{2,78} = 18.84$, $p < 0.001$, $\eta_p^2 = 0.33$) and while decoding was also higher for HC than for LC trials ($F_{1,39} = 11.64$, $p = 0.002$, $\eta_p^2 = 0.23$, **Fig 3C**), the interaction between these 2 factors was not robust, although numerically, decoding was higher in HC compared to LC trials, in particular at the later versus early stage ($F_{2,78} = 2.81$, $p = 0.07$, $\eta_p^2 = 0.07$, $BF_{excl} = 1.96$). A series of planned simple comparisons showed that the flip from negative to positive decoding was stronger for high confidence decisions (results are summarized in **Fig 3E**). The below-chance decoding during the 2 early diagonal decoding time-windows on incorrect decisions indicated that the classifiers were picking up the presented stimulus category during sensory stages of visual information processing. At the earliest time-windows (80 to 130 ms), we did not find evidence that decoding was modulated by confidence (HC versus LC; $t_{39} = -0.28$, $p = 0.78$, $d = -0.05$, $BF_{01} = 5.64$) and indeed, decoding was below chance-level for both confidence levels (LC: $t_{39} = -5.83$, $p < 0.001$, $d = -0.92$; HC: $t_{39} = -3.38$, $p = 0.002$, $d = -0.53$). During the following mid-stage (150 to 200 ms), classifiers also mainly decoded the veridical stimulus category, but interestingly in particular when participants reported low confidence in their decision (LC: $t_{39} = -5.05$, $p < 0.001$, $d = -0.8$; HC: $t_{39} = -0.35$, $p = 0.73$, $d = -0.06$, $BF_{01} = 5.54$; LC versus HC: $t_{39} = -2.95$, $p = 0.005$, $d = -0.47$). Then finally, during the late processing stage (350 to 500 ms), patterns of neural activity represented the incorrectly reported stimulus category, but in particular when participants where highly confident in their decision, so when the illusion was strongest (HC: $t_{39} = 3.53$, $p < 0.001$, $d = 0.56$; LC: $t_{39} = 0.44$, $p = 0.67$, $d = -0.07$, $BF_{01} = 5.36$, LC versus HC: $t_{39} = -3.28$, $p = 0.002$, $d = -0.52$). These results show a reversal from the presented stimulus category to the reported category for incorrect decisions, especially for high confidence decisions.

## Off-diagonal decoding based on the decision localizer

Next, we examined decoding of the reported stimulus category off-diagonally, training on the N170 time-window (150 to 200 ms) using the decision localizer task data, separately for correct

and incorrect decisions (only 1 stage). First, on correct decisions, decoding of the reported stimulus category was robust for both confidence levels (all $p$'s $< 0.001$), although decoding was higher for HC than LC decisions ($t_{39} = -3.98$, $p = <0.001$, d = −0.63; **Fig 3D and 3E**, right panel), suggesting that the veridical/perceived stimulus category was maintained perceptually. On incorrect decisions, however, what was perceptually represented off-diagonally was not the presented stimulus, but the misreported stimulus category, and this effect was stronger for high confident than low confident incorrect decisions (HC: $t_{39} = 3.86$, $p < 0.001$, d = 0.61; LC: $t_{39} = 1.41$, $p = 0.17$, d = 0.22, $BF_{01} = 2.35$; LC versus HC: $t_{39} = -3.06$, $p = 0.004$, d = −0.48; **Fig 3E**). Thus, based on the transient decoding peak observed at early processing stages (150 to 200 ms in the decision localizer), the erroneous and misreported stimulus category was represented in a persistent perceptual format later in time, despite never being presented on the screen and never being neurally represented at earlier time points (**Fig 3D**). Statistical tests across time revealed that the incorrect off-diagonal decoding trace emerged relatively early, starting at approximately 213 ms and peaking at 236 ms after stimulus presentation (**Fig 3D**, horizontal blue colored bars reflect significant time points after cluster-based permutation testing). Although we observed a perceptual representation of an incorrectly reported category, concluding that errors in decision-making in this task have a true perceptual origin (in the case confidence is high), based on the current results may be premature, because the decoded signal could reflect a combination of perceptual and decision-related processes [20,52]. Therefore, we further examined this issue when training our classifiers on the sensory tuned localizer.

## The time course of category representations: Sensory localizer

To explore to what extent errors in decision-making have a true perceptual origin, we performed the same set of analysis, but now while training the classifier on the sensory localizer task (**Fig 4A**). In this task, participants performed a simple change detection task on the color of the fixation cross making the face/house images fully task irrelevant. Because the sensory localizer task does not have a decision component to the face/house images, it is uniquely sensitive to sensory features. Therefore, we expected that the late square-shaped on-diagonal decoding pattern, possibly related to global ignition [34,37,50], would be strongly reduced or even disappear when cross classifying between the sensory localizer and the main perceptual decision task (**Fig 4B**). Previous studies have shown that this late square-shaped decoding pattern is likely related to decision processes arising after perceptual analysis, which disappear when visual input is task irrelevant [18–20,53]. On the other hand, because decoding is still performed on the face/house images, the sensory stages should be relatively unaffected by this task relevance manipulation [18,49]. In this Results section, we focus on those aspects of decoding that are informative for addressing the question whether decision errors have a perceptual origin, but note that all other effects were similar to the analyses reported in the "decision localizer" section and the details thereof can be found in the Supporting information (including all the ANOVA's).

For correct decisions, decoding was above chance for both confidence levels (all $p$'s $< 0.001$, see **Fig 4C**) in the early and mid-stage, but we did not find evidence for decoding at the late diagonal stage (LC: $t_{39} = 1.104$, $p = 0.28$, d = 0.18, $BF_{01} = 3.33$; HC: $t_{39} = 1.34$, $p = 0.19$, d = 0.21, $BF_{01} = 2.55$). Thus, as anticipated, the late on-diagonal 350 to 500 ms stage was not robust when training on the sensory localizer and classifying on the main discrimination task, whereas all other stages of visual processing remained relatively intact (**Fig 4C**). During the perceptual maintenance stage, the activity patterns reflected the veridical/reported stimulus category (both $p$'s $< 0.001$, LC versus HC: $t_{39} = -0.98$, $p = 0.34$, d = −0.15, $BF_{01} = 3.76$, **Fig 4D**).

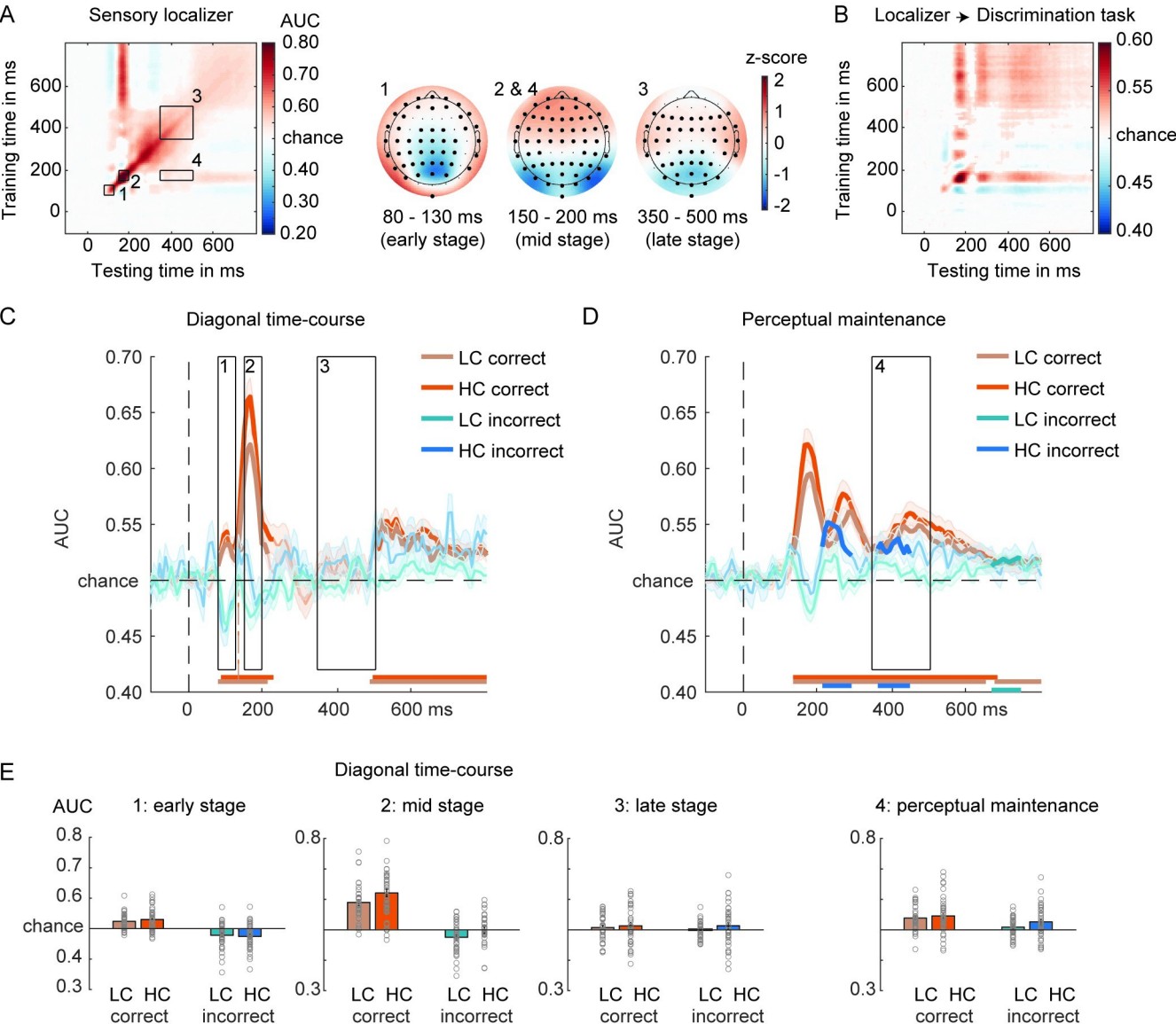

**Fig 4. Face/house classification based on the sensory localizer.** (**A**) GAT matrix for the sensory localizer (all electrodes) and ROI marked by inset black boxes (numbers 1–4). On the right, covariance/class separability maps for each diagonal ROI, indicating the underlying data contributing to the classification performance. Highlighted channels significantly contributed to the classification performance after cluster-based permutation testing over channels (corrected for multiple comparisons). Covariance/class separability maps 2 and 4 are the same because decoding scores within insets 2 and 4 were obtained using the same training time-window. (**B**) Classifiers trained on the sensory localizer applied to the main discrimination task (cross-task validation procedure, all electrodes). (**C**) On-diagonal and (**D**) off-diagonal time courses of correct and incorrect decisions, separated by confidence. Black boxes indicate the time-windows (early, mid, late) used for averaging the AUC scores. Colored horizontal lines indicate periods of significant decoding with respect to chance. Longer-lasting sensory traces are derived by training a classifier on the time-window 150–200 ms and testing it across the entire time-window (panel D). Classification performance was evaluated at each time point using cluster-based permutation testing (two-tailed cluster-permutation, alpha $p < 0.05$, cluster alpha $p < 0.05$, N permutations = 1,000). (**E**) Bar plots show average AUC values for the TOI highlighted in panels (C) and (D). The underlying data and scripts supporting this figure can be found on FigShare (https://doi.org/10.21942/uva.c.6265233.v1). AUC, area under the curve; GAT, generalization across time; HC, high confidence; LC, low confidence; ROI, regions of interest; TOI, time-windows of interest.

For incorrect decisions, the early and mid-stage were also similar as compared to when the classifiers were trained on the decision localizer (early stage: LC: $t_{39} = -3.4$, $p = 0.002$, d = $-0.54$; HC: $t_{39} = -3.91$, $p < 0.001$, d = $-0.62$; mid-stage: LC: $t_{39} = -3.04$, $p = 0.004$, d = $-0.48$; HC: $t_{39} = -0.11$, $p = 0.91$, d = $-0.02$, $BF_{01} = 5.83$). Specifically, we observed negative decoding

for both early stages (again, except for HC trials in the mid-stage), reflecting the processing of the veridical stimulus input. Crucially, also here, the late-stage decoding disappeared (LC: $t_{39}$ = 0.16, $p$ = 0.88, d = −0.03, $BF_{01}$ = 5.8; HC: $t_{39}$ = 1.27, $p$ = 0.21, d = 0.2, $BF_{01}$ = 2.8, **Fig 4C**). However, most importantly, during the off-diagonal perceptual maintenance stage using the classifier trained on the sensory localizer data, we could still decode the incorrectly reported stimulus category, in particular when participants reported high confidence in their decision (LC: $t_{39}$ = 1.78, $p$ = 0.08, d = 0.28; $BF_{01}$ = 1.4; HC: $t_{39}$ = 3.03, $p$ = 0.004, d = 0.48, $BF_{01}$ = 0.12; LC versus HC: $t_{39}$ = −1.94, $p$ = 0.06, d = −0.31, $BF_{01}$ = 1.07, **Fig 4D and 4E**). Again, here as well, cluster-based testing across the entire time-window revealed that the perceptual representation of the incorrect stimulus spontaneously emerged right after the 150 to 200 ms time-window most strongly representing face/house differentiation (see **Fig 4D**). This finding shows that misreported decisions may be due to true perceptual illusions.

## Stimulus orientation decoding

Our results so far show that initial information processing stages are stimulus related, whereas later off-diagonal decoding captures perceptual aspects of decision errors. To show that these effects are specific for task-relevant features of the images (face/house category), we additionally focused on stimulus orientation decoding as a function of correctness and confidence. Stimulus orientation was a task-irrelevant feature of the stimuli and previous studies have shown that it can be decoded already approximately 100 ms post-stimulus [50]. A single isolated decoding peak within the 30 to 130 ms post-stimulus was observed both for within-task decoding (**Fig 5A**, 10-fold decoding) as well as cross-task classification (decision classifiers ➔ main discrimination task, **Fig 5B and 5C**). Decoding strength was similar for correct and incorrect decisions ($F_{1,39}$ = 0.06, $p$ = 0.82, $\eta_p^2$ = 0.001, $BF_{01}$ = 5.7) and more evidence was observed for the absence of an effect. We also did not find evidence that confidence modulated orientation decoding in any way ($F_{1,39}$ = 0.54, $p$ = 0.47, $\eta_p^2$ = 0.014, $BF_{01}$ = 4.77; interaction correctness × confidence: $F_{1,39}$ = 0.18, $p$ = 0.67, $\eta_p^2$ = 0.005, $BF_{excl}$ = 4.22).

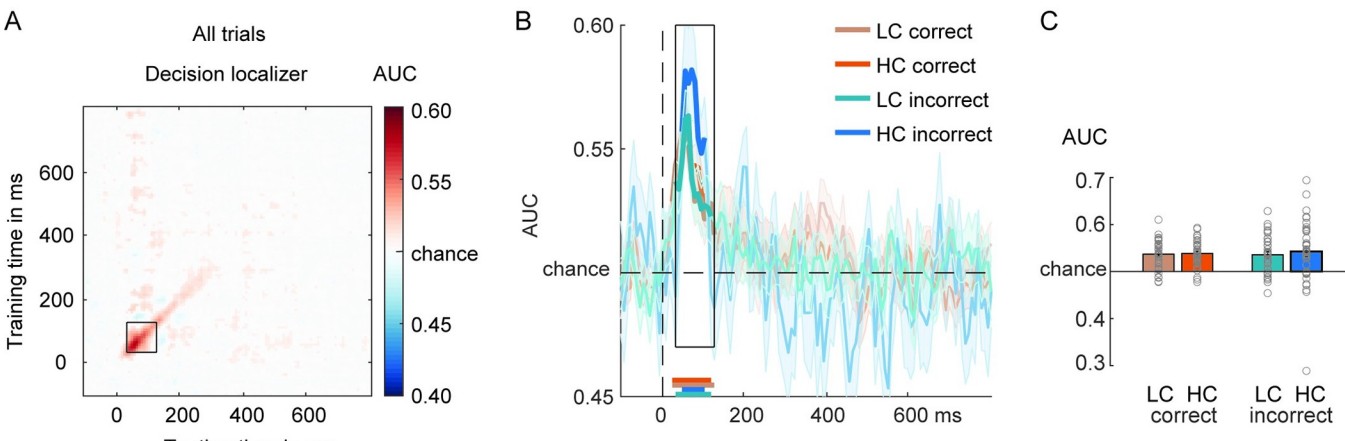

**Fig 5. Tilt (right/left) classification based on decision localizer.** (**A**) GAT matrix of the decision localizer data for arbitrating left/right tilted stimuli (irrespective of face/house category, based on occipital–parietal channels). The black square-shaped inset indicates the time-window used for averaging AUC decoding scores (30–130 ms). (**B**) On-diagonal time courses for correct and incorrect decisions separated by confidence. Colored horizontal lines indicate periods of significant decoding with respect to chance. Classification performance was evaluated at each time point using cluster-based permutation testing (two-tailed cluster-permutation, alpha $p$ < 0.05, cluster alpha $p$ < 0.05, N permutations = 1,000). (**C**) Bar plots show average AUC values for the time-window of interest highlighted in panels (A) and (B). The underlying data and scripts supporting this figure can be found on FigShare (https://doi.org/10.21942/uva.c. 6265233.v1). AUC, area under the curve; GAT, generalization across time; HC, high confidence; LC, low confidence.

## Discussion

The goal of this study was to investigate whether errors during perceptual decision-making have a sensory or a decisional origin. We isolated sensory versus decision-related neural processes by using 2 distinct functional localizers, differently tuned to these processes [20]. Replicating a wealth of previous studies, the earliest neural responses reflected the physical properties of the presented stimulus [51–55]. The crucial finding reported here is however that when participants made an incorrect perceptual decision with high confidence, early stages of visual information processing (<~200 ms post-stimulus) still represented object features of the presented stimulus (e.g., "faceness" in the case a face stimulus was presented), while later stages represented the illusory perceptual properties of the reported stimulus category (e.g., "houseness" in case a face was presented, but a house response was given). More importantly, at least 3 findings reported here demonstrate the sensory and true illusory nature underlying such high confident erroneous perceptual decisions and thereby go beyond traditional interpretations associating late neural processing with largely stimulus-driven perception.

First, persistent sensory-based decoding across time (perceptual maintenance, off-diagonal decoding) on incorrect trials was observed when training a classifier on the N170-like processing stage, a peak in the EEG signal highly associated with face perception [39,44,49–51,56–58]. Whereas low-level visual features, such as stimulus orientation and position, drive early-latency decoding (<100 ms post-stimulus), slightly later in time decoding has been shown to reflect the integration of sensory features into coherent objects and object categories (as early as approximately 150 ms post-stimulus, [8,19,50,51,59]). Regarding face perception, a recent MEG study showed that decoding of face representations peaked at approximately 160 ms post-stimulus, and this signal correlated with subjective ratings of "faceness" of inanimate objects [57]. By combining MEG and functional magnetic resonance imaging (fMRI) measures, it was shown that these face responses likely originated from the FFA in ventral temporal cortex (see also [49,56]). This mid latency face-like component has been shown to be relatively unchanged when object categories are task irrelevant [18]. Others have also provided evidence for strong occipital contributions to the decoding of perceptually integrated objects and object categories during similar early latencies [34,60]. These findings highlight that the mid latency (150 to 200 ms) decoding stage indexes perceptually coherent category representations, likely originating from (higher level) visual cortices. A novel line of evidence based on decoding and generalization across time [61] has shown that these category representations are decodable for much longer periods of time than originally anticipated, as reflected in off-diagonal decoding patterns, focused on in this work [33–35,53]. Also in the data presented here, sensory representations were decodable for relatively long periods of time (see **Figs 3 and 4**) [35].

A second reason why erroneous decisions in our task are likely caused by true perceptual illusions is that long-lasting late sensory representations were observed even when classifiers were trained on the sensory-tuned localizer task (perceptual maintenance, **Fig 4D**). Sensory-tuned classifiers were trained on a task in which face/house images were fully task irrelevant and participants focused their attention on detecting small changes in the color of the fixation cross. Therefore, this classifier was only able to pick up (unattended) sensory features of the images that arbitrate between faces and houses, in the absence of most post-perceptual and decision-related processes [18–20,26]. That we were successful in isolating sensory features was evident in the strong decrease of the late on-diagonal "square-shaped" decoding profile while training on the sensory localizer task. Another approach to isolate neural signals associated with perceptual experience from decision- and report-related processes is the development of no-report paradigms. In such paradigms, observers may be aware or unaware of presented stimuli, but in all cases do not make perceptual judgments about them. The resulting

contrasts (aware versus unaware) is then supposed to isolate perceptual experience, limiting the influence of post-perceptual processes. Also in this line of work, it has been shown that the majority of the high-level activations in frontal and parietal cortex disappear (similar to our late square-shaped decoding), and hence, that such activation patterns are mostly associated with post-perceptual processes and much less so with perceptual experience itself [21–26].

Third and finally, the flip from veridical stimulus-related processing relatively early in time towards illusion-based (or report-based) processing later in time was only observed for high confidence decisions and not for low confidence decisions. In other words, the emergence of these perceptual representations in the EEG signal depends on the strength of the perceptual illusion, assuming that the perceptual illusion was strong for high confidence (inaccurate) decisions and weak for low confidence decisions (see [62] for a way to explore "partial errors" by inspecting the electromyography signal on correct responses). Low confidence decision errors were likely the result of guessing [7]. In a related fMRI study, Summerfield and colleagues have shown that misreporting a house for a face in a challenging discrimination task similar to ours, is accompanied by increased fMRI BOLD activity in the FFA, but not in other face responsive regions such as the occipital face area (OFA) [7]. This suggests that the FFA is associated with illusory perception of faces, whereas earlier face processing regions in the cortical hierarchy are not, and reflect the presented or veridical stimulus category [7,63,64]. Although in this fMRI study, discrimination responses were also given with high or low confidence, unfortunately too few high confidence error trials could be obtained to perform the crucial analyses that test for confidence modulations on decision errors. Furthermore, in this study, image contrast varied across trials, and therefore correctness, confidence, and image contrast co-varied, which complicated the separation between perceptual effects and effects of bottom-up input strength (e.g., decision errors were also often low contrast trials). However, together these results highlight that relatively early stages in object processing, both in time as well as in the cortical hierarchy, are linked to the image input, whereas later stages are more internally driven or perceptual in nature. Intriguingly, our results also show that emerging perceptual representations do not necessarily have to follow from bottom-up sensory processes. When participants misreported the presented stimulus category (on incorrect trials), we found no N170-like decoding pattern for high confidence decisions, but we did observe the emergence of perceptual representations of the reported category later in time (**Fig 3B**). Thus, illusory category-specific perceptual signatures emerged without any clear bottom-up sensory evidence for this percept.

Although these results reveal that high confidence errors are associated with neural representations indicative of true perceptual illusions, they leave open the question what drives these reversals in representations from the veridical to the misreported stimulus category. There are several likely candidate mechanisms, which future studies should aim to arbitrate between. One intuitive candidate may be decision-related feedback to visual cortex. The frontoparietal network involved in perceptual decision-making has strong feedback connections to sensory regions and can continuously inform sensory regions about the unfolding decision variables through feedback connections [65,66]. In general, activity observed in sensory regions can thus be a mixture of both feedforward and feedback processes, especially at longer latencies (>~100 ms) ([39], but see [40]), making it difficult to disentangle sensory and post-sensory or decision processes even at the level of single sensory neurons [67–74]. fMRI studies in humans have for example shown that early visual cortex activity may reflect a combination of stimulus-related feedforward activity and post-sensory decision-related feedback [75] or top-down processes such as task set and expectations [76]. The observation that we only observed the flip form veridical to the reported perceptual representations in high confidence trials runs contrary to the interpretation that decision-related feedback may be the best

explanation of the observed data patterns. If these effects were purely decision-driven, we should have observed similar decoding patterns for low confidence decisions, although we cannot fully rule out the possibility that these sensory effects were caused by decision-related feedback.

Another possibility is that, because the bottom-up visual input is impoverished due to strong masking, higher-order areas may incorrectly explain the noisy bottom-up signals. Previous evidence suggests that sensory templates can be implemented in a top-down manner even in anticipation of sensory stimulation [77–79] and a body of work shows that the FFA is sensitive to top-down factors, such as task-relevance, expectations, and context [76]. For example, seeing a face in ambiguous Mooney images [80] or face-like inanimate objects, the phenomenon know as *pareidolia* [57], leads to stronger FFA responses than when no face is perceived, despite very similar bottom-up input. That the FFA is sensitive to context is also illustrated by the observation that the FFA activates to simple oval shapes when the context suggests this shape might be a face [81]. Furthermore, Summerfield and colleagues showed effects of task set on object processing [82]. They administered a task in which in 1 block of trials participants had to detect faces among faces, houses, and cars, and in another block, they had to detect houses, among the same 3 categories of stimuli. Thereby the authors manipulated the top-down "perceptual set" while keeping the bottom-up input the same across blocks. When comparing the overall activity in the FFA on blocks in which faces had to be detected versus houses had to be detected, increased FFA activity was observed. The authors concluded that top-down signals from frontal cortices could sensitize visual regions responsible for collecting evidence about the presence of faces. Although we did not directly manipulate context or expectations in our design (faces and house were equiprobable), participants may have developed expectancies about image likelihood over time, which may have then led to explaining the noisy sensory input, probably at the level of object-selective cortex, in an erroneous way [7]. Another related and interesting finding has been reported by Tu and colleagues [8]. In their task, participants had to discriminate faces from houses or cars while EEG and fMRI were measured simultaneously. As the authors have observed in different task contexts [18,83], there were both early (approximately 200 ms after stimulus) and late (approximately 500 ms after stimulus) EEG components that discriminated between stimulus categories. Interestingly, in behavior, participants had an overall face bias: they often misreported nonfaces as faces. By combining EEG with fMRI, the authors showed that individual differences in the strength of this face-bias correlated across participants with the strength of the interaction between "early" (e.g., FFA, parahipocampal place area, parietal regions) and "late" neural networks (primarily frontal regions, e.g., anterior cingulate cortex). Therefore, the strength of a face decision bias may depend on the degree of top-down predictive modulations from frontal to sensory cortices. Because no single trial confidence reports were obtained in that study, the authors could not relate their neural data and categorical responses to introspective confidence in the observers' decisions. Our study therefore extends this previous work by showing that (at least) high confidence decision errors have a perceptual origin. In general, together these studies suggest that both perceptual biases and decision biases depend on interactions between top-down and bottom-up processes (see also [9,12,14,84] for ways aiming to behaviorally disentangle sensory from decision biases).

Predictions, however, do not have to be strategic or conscious, but may fluctuate naturally over time in neural activity, e.g., in object selective cortex [85]. It has been argued that ongoing fluctuations of neural activity are not random (i.e., stochastic noise), but may contain content-specific information, for example, associated with previously experienced stimuli and (perceptual) learning [86–93]. It has, for example, been shown recently that when patterns of pre-stimulus neural activity measured with magnetoencephalography matched the stimulus category

that was later presented, which had to be discriminated from other object categories by human participants, perceptual sensitivity was improved [85]. Thus, another possibility is that, in the face of uncertainty, when sensory input is weak or ambiguous, the system may settle on the most likely interpretation of the sensory input, integrating the sensory evidence and the ongoing activity present at the moment the stimulus travels through the cortical hierarchy. In our study, when perceptual decisions were incorrect, but made with high confidence, the perceptual system may have settled in a state coding for the specific stimulus category that was perceived, leading to a true illusion in perception. Yet, in low confidence decision error trials, the perceptual system may have not settled on any of the 2 stimulus categories, as strikingly reflected in the absence of significant decoding for either the presented or the reported object category.

To conclude, we find that object categorization errors are associated with a quick reversal in sensory representation from the veridical, presented stimulus category to the reported stimulus category, but only for decision made with high confidence. This finding shows that high-confidence decision errors are caused by true illusions in perception.

## Methods

### Ethics statement

Participants provided written informed consent prior to the start of the experiment and were tested following a protocol approved by the ethical committee of the Department of Psychology of the University of Amsterdam (project number: 2019-BC-10724).

### Participants

A total of 44 participants (33 female, mean age = 23.34; SD = 5.68) from the University of Amsterdam, all right-handed, with reported normal or corrected-to-normal vision and no history of a psychiatric or neurological disorder, were tested in this study. Participants received research credits or money (10 euros per hour) for compensation. Forty participants, all between the age of 18 and 35, completed 2 experimental sessions of this study and comprised the final sample included in all reported analyses (30 female, mean age = 22.83; SD = 3.31). The remaining 4 participants were identified based on their data from session 1, after which they were excluded from further participation/analyses. One participant was excluded from the final analysis because of surpassing the age criterion (age > 35), while the other 3 participants were excluded due to the around-chance performance in 1 or both localizer tasks in the first session.

### Materials

All tasks used in the current study were developed and executed using Matlab 8 and Psychtoolbox-3 software within a Matlab environment (Mathworks, RRID:SCR_001622). Stimuli were presented on 1,920 × 1,080 pixels BenQ XL2420Z LED monitor at a 120-Hz refresh rate on a black (RGB: [0 0 0], ± 3 cd/m$^2$) background and were viewed with a distance of 90 cm from the monitor using a chin rest.

### Procedure and stimuli

The experiment consisted of 2 sessions, each approximately 3 h long and scheduled on 2 different days for each participant. In the first session, participants completed a change detection task and a discrimination task while we measured their brain activity using EEG. In the change detection task, participants focused on the central fixation cross and reported its color change

whenever it changed from red to light red, while a stream of brief house and face images was shown simultaneously on the screen (**S1A Fig**). In the discrimination task, in every trial, participants saw a masked image of a face or a house (**S1A Fig**). At the end of each trial, they were asked to indicate which stimulus category they perceived. The discrimination task administered in the second session was highly similar (see the details of the design below), but in addition to reporting which stimulus category they perceived, participants also provided confidence ratings regarding their decision.

MVPA classifiers were trained on EEG data recorded during the first session, separately for the change detection task ("sensory" localizer) and the discrimination task ("decision" localizer), which were then applied to the discrimination task EEG data recorded in the second session.

*Session 1*: *Change detection task (sensory localizer)*. In this task, participants focused their attention on a centrally presented fixation cross, which was superimposed on a rapidly changing sequence of house and face images (**S1A Fig**). The fixation cross remained present on the screen throughout a block of trials. Participants' task was to monitor the fixation cross and report its brief color change, from red (RBG: [255 0 0]) to a lighter shade of red (RBG: [125 0 0]) for 100 ms on 20% of randomly chosen trials. Participants were instructed to press the spacebar whenever they noticed the color change of the fixation cross. Note that this change detection task adds an orienting response to the fixation cross on some trials, which was however unrelated to, and not predictive of, the occurrence of the task-irrelevant face or house stimuli.

Stimuli set consisted of 180 unique houses and 180 unique faces (90 male and 90 female). Face and house images were gray scale stimuli, obtained from [33]. Face and house stimuli were equated for spatial frequency and luminance (for details, see [33]). All stimuli subtended $16 \times 20°$ visual angle, were presented centrally on a black background, and were tilted to the right or left at a 5° or 355° angle, respectively. Each image was shown for 100 ms with the ISI that was jittered between 1,200 and 1,400 ms. The task-relevant color change of the fixation cross could occur only in the ISI, at a randomly determined moment 15 ms after the start of the ISI and 130 ms before its end. Participants were instructed not to pay attention to images of houses and faces while maintaining their fixation at the center of the screen. Images of houses and faces were thus task irrelevant, although they were concurrently processed visually.

By ensuring that participants' attention was focused on the centrally presented task, the aim was to minimize the possibility that MVPA classifiers were impacted by systematic eye movements that could be strategically deployed to discriminate between stimuli classes [94]. For example, a consistent eye movement towards the top of the stimulus to detect face-defining features (e.g., eyes) could alone drive multivariate differences between houses and faces stimuli.

In total, 200 images of houses and faces (100 of each category, half tilted to the right and half tilted to the left) were shown in each of 6 experimental blocks. Before the start of the first experimental block, each participant completed 1 practice block of 100 trials in order to get familiar with the task.

*Session 1*: *Discrimination task (decision localizer)*. The task design was highly similar to the task developed by Weaver and colleagues [33]. An overview of the trial procedure can be found in Fig 1A (and **S1B Fig**). Each trial started with a fixation dot that remained on the screen throughout the duration of the trial. After a fixation-only interval that was jittered between 600 and 1,000 ms, a scrambled mask stimulus appeared on the screen for 50 ms, followed by a target face or a house stimulus, shown for 80 ms. The target image was followed by another 50 ms-long scrambled mask stimulus. A response screen was presented for 1,000 ms immediately after the offset of the second mask stimulus, during which participants needed to give a speeded response indicating whether they saw a house or a face, using a left-handed

("z") or right-handed ("m") keyboard response. Stimulus-response mappings were randomized across blocks of trials to prevent motor response preparation before the response screen was shown. The correct stimulus-response mapping (e.g., a left button press for a face and a right button press for a house stimulus) was presented at the beginning of each block. Furthermore, letters F and H (approximate size 4 × 4˚), for faces and houses, respectively, were shown during the entire trial in the right and left upper corner (centered approximately 20 × 24˚ from fixation), or vice versa, depending on the block, as a reminder of the response mapping to the button press.

Face, house, and mask stimuli were obtained from the study by Weaver and colleagues [33]. Stimuli set consisted of 180 unique houses and 180 unique faces of which 90 faces were male and 90 were female. Face and house stimuli were equated for spatial frequency and luminance (for details see [33]). Visual masks were selected from 900 scrambled face and house images (parsed into 12 × 15 tiles and randomly shuffled) that had been made transparent and superimposed. All stimuli were gray scale, subtended 16 × 20˚ visual angle, and were presented centrally on a black background. Target stimuli and masks were shown tilted to the right or left, at a 5˚ or 355˚ angle, respectively.

Participants completed 14 experimental blocks containing 64 trials each, 896 trials in total. In each block, an equal number of faces and houses were shown, half of which were tilted to the right and half to the left. Before the first experimental block, each participant completed 1 practice block of 64 trials to get familiar with the task.

*Session 2*: *Discrimination task*. The task and procedure of the second session discrimination task was highly similar to the discrimination task administered in the first session (cf. [33]). Here, participants again viewed a rapid stream of house and face images and their task was to report at the end of the trial whether they had seen a house or a face. Right (5˚) or left (355˚) tilted house and face images were preceded and followed by visual masks, which were also tilted in the same direction. The trial procedure was identical to the discrimination task of session 1, except that the target face or house was presented for 20 ms or 30 ms. The duration of target stimuli was determined per participant depending on the discrimination performance during practice blocks, with the aim of achieving approximately 65% discrimination accuracy. Specifically, if a participant scored 60% or higher in correctly discriminating target stimuli when they were shown for 20 ms, this timing was used as the target duration for the remaining experimental blocks. Otherwise, target stimuli were shown for 30 ms.

After a 50 ms-long post-mask, a response screen was presented for 1,300 ms, during which participants needed to give a speeded response indicating whether they saw a house or a face using a left-handed ("z") or right-handed ("m") keyboard response. Additionally, following the discrimination response, participants needed to indicate their confidence in the accuracy of their discrimination response using a 4-point scale (1 –unsure, 4 –sure). The next trial began after the response had been given or after a 3-s timeout if no response was recorded. An equal number of right- and left-tilted faces and houses were shown in each block.

The discrimination task started with 4 practice blocks of 84 trials each, administered to familiarize participants with the task and to get an indication of target presentation time for the remainder of the task. This was done to ensure that the stimulus presentation time was sufficiently long, so that accuracy scores during the practice averaged to around 65%. Following the practice, each participant completed 16 experimental blocks of 84 trials each.

## EEG measurements and preprocessing

The electroencephalogram (EEG) and electrooculogram (EOG) were recorded using the Biosemi Functional Two system (Biosemi.com). A total of 64 sintered AG/AgCl electrodes were

positioned according to the 64 standard international 10/20 system, 6 external electrodes were placed on the earlobes and around the eye. The vertical EOG (VEOG) was recorded from 2 external electrodes located above and below the right eye. The horizontal EOG (HEOG) was recorded from 2 external electrodes located next to the external canthi of the eyes. The VEOG was used to detect eyeblinks and the HEOG was used to detect horizontal eye movements. Electrophysiological signals were digitized at a sampling rate of 512 Hz.

EEG data was preprocessed and cleaned before further analysis using custom scripts, the EEGLAB toolbox (v2019_1), and the Amsterdam Decoding and Modeling toolbox (ADAM; [95]). EEG data was re-referenced to the average of the earlobes, high-pass filtered at 0.5 Hz and low-pass filtered at 40 Hz, in separate steps as recommended by EEGlab. Note that a high-pass filter may distort the temporal estimates of the EEG signal [96,97], which was not the case in our dataset (see **S3 Fig** illustrating this). The continuous data was then epoched from −500 ms to 2,000 ms around stimulus onset. Trials containing jump artifacts were removed from the data using an adapted version of ft_artifact_zvalue muscle artifact detection function from the FieldTrip toolbox. Eye blink artifacts were removed from the data using a standard regression-based algorithm. A baseline correction in the 200-ms pre-stimulus onset interval was performed. Data were then downsampled to 128 Hz.

## Statistical analyses

All analyses were done using custom scripts, the EEGLAB toolbox (v2019_1), the Amsterdam Decoding and Modeling toolbox (ADAM; [95]), and JASP software. We took a frequentist statistical approach and chose an alpha of $p < 0.05$ as the threshold for significance as is typical, to guard against false positives. Note, however, that this threshold is relatively arbitrary, and that frequentist statistics cannot provide evidence for the null hypothesis (absence of an effect) [98]. Therefore, we also report effect sizes, and in case of nonsignificant effects as indicated by $p > 0.05$, followed up with a Bayesian equivalent of the same test in order to quantify the strength of evidence for the null hypothesis ($H_0$). By convention, Bayes factors from 1 to 3 were considered as anecdotal, 3 to 10 as substantial, and those above 10 as strong evidence in favor of $H_0$.

## Behavior

Trials in which participants made a discrimination response <200 ms after the stimulus presentation (3.39%) or gave no response at all (1.86%) were discarded from the analysis. To evaluate participants' behavioral performance, we computed d-prime (d', Type-I sensitivity), a measure of perceptual sensitivity to presented stimuli, separately for each confidence level and analyzed using a one-way repeated measures ANOVA with reported confidence ratings [1–4] as within-participant factor.

## Decoding analyses

MVPA was applied to EEG data in order to decode patterns of neural activity specific to house and face stimuli. First, to test if we could decode category-specific neural representations in the localizer tasks, we used a 10-fold cross validation scheme. In this procedure, separately per localizer task, after randomizing the order of trials, the dataset was split into 10 equally sized folds. Equal number of house and face stimuli were present in each fold. We then trained a linear discriminant classifier (LDA) [27] to differentiate between house and face images using 9 folds and tested its performance on the remaining fold (using the standard Matlab function fitcdiscr). This was repeated 10 times, until all data were tested exactly once. Classification performance was evaluated for each participant separately by computing the area under the curve

(AUC), which indicates the degree of separability between classes of the receiver operating characteristic (ROC) curve. First, we computed the proportion of correct classifications for each stimulus category, after which the scores were averaged across stimulus categories and over the 10 folds. Prior to training and testing procedures, EEG data was downsampled to 128 Hz and epochs were shortened to −100 to 800 ms, centered on the target stimulus, in order to decrease the computational time needed for MVPA.

Decoding analyses were carried out using EEG data recorded at all electrodes and a set of occipital–parietal electrodes [33]. The occipital–parietal set of electrodes, chosen so that it captures early visual "N170-like" response to houses and faces, consisted of Iz, Oz, O1, O2, POz, PO3, PO4, PO7, PO8, Pz, P1, P2, P3, P4, P5, P6, P7, P8, P9, and P10 electrodes.

To test our specific research questions, we used the cross-task validation scheme to evaluate the performance of each classifier (trained on either the sensory or the decision localizer task data) in differentiating between stimulus classes in the main decision task. We applied the same linear discriminant classifier to raw EEG data from the localizer task using voltages at each time sample to train the classifier, which was then applied to the main discrimination task data. This was done for separately for the sensory and decision localizer tasks. Following this procedure, we could examine whether and when house- and face-specific neural representations were associated with participants' confidence in their perceptual decision, and whether perceptual decisions, in particular when incorrect, rely on the category-specific pattern of activation that is specific for the reported percept, or alternatively, representations are specific for the presented category. Moreover, critically, we could isolate sensory versus decision-related neural processes as the 2 localizers were differently tuned to these processes.

To examine the representational nature of decision errors, and to uncover how category-specific representations evolve across time, and across distinct processing stages, we used the GAT approach in applying the pattern classifiers [61]. Specifically, a classifier trained on a specific time point was tested on that time point as well as on all other time points. The resulting GAT matrix (training time x testing time) thus reveals dynamic changes of neural representations across time. For example, a classifier trained to decode between house and face images at 170 ms can generalize to a wider time-window, e.g., 170 to 350 ms, in which case it would indicate that the early neural representation was maintained in time. This approach is thus informative of how neural representations change across different stages of visual information processing, also permitting us to examine when in time neural representations differ between the perceived stimulus classes and depending on the correctness and confidence therein.

Decoding analyses were performed separately for correct and incorrect trials as a function of 2 confidence levels. Trials in which participants reported confidence ratings 1 and 2 were aggregated into "low" confidence trials, and those on which ratings 3 and 4 were given into "high" confidence trials. To illustrate, when decoding was performed for incorrect high and low confidence trials, classifiers were trained on the presented image category of the localizer task but were tested using trials in which participants reported seeing incorrect stimulus category, followed by high versus low confidence rating. Besides the stimulus category, we also decoded its orientation, i.e., whether a stimulus was tilted to the right or to the left, separately for correct and incorrect trials, as a function of confidence level. This analysis served 2 purposes. First, we wanted to verify that stimulus processing from the very bottom-up input did not differ between high versus low confidence trials. Low-level visual features such as orientation should be encoded in early brain activity in a bottom-up manner [50], but were not expected to be modulated by confidence due to the early timing of that processing stage. Second, the stimulus orientation was a task-irrelevant feature that participants did not need to do anything with and was therefore useful to tests whether task-irrelevant features are processed differently on high versus low confidence trials. To this end, classifiers were trained to

distinguish between right versus left tilt of the stimulus using localizer data and then tested on right versus left images orientations using trials in which the correct versus incorrect stimulus category was reported.

Statistical analyses were performed on average classification scores (AUC) in time-windows that we preselected based on previous empirical work. Typically, studies report 2 processing stages using the GAT approach: an early (<250 to 300 ms) cluster of diagonal decoding reflecting initial sensory processes and a late processing stage (>300 ms), which was found to associate with conscious report [33,34,37]. Additionally, some studies have reported a third stage of visual information processing, starting at early latencies and generalizing off-diagonal, presumably reflecting maintenance of early sensory stimulus representations [33,34,53]. Following this body of work, and the study by Weaver and colleagues [33], who used an almost identical task design as in the present study, we focused our statistical analyses on 2 decoding clusters on-diagonal (classifiers were trained and tested on the same time point) and 1 decoding cluster off-diagonal (training and testing were done on different time points). The 2 on-diagonal clusters were 150 to 200 ms and 350 to 550 ms. The off-diagonal time-window spanned from 150 to 200 ms training time to 350 to 550 ms testing time (see below for more details). Finally, although decoding was pronounced during 3 time-windows observed previously by Weaver and colleagues [33], it was also observed at an even earlier time-window (80 to 130 ms) with a prominent occipital–parietal topography (**Fig 2A**, inset 1, early-stage topography). Because this early effect was absent in the same task when images where presented vertically [33], this early peak likely reflects differences in the orientations of the images presented, which we varied systematically across trials. This early window was also used as an event of interest in this study.

To inspect the pattern of neural activity that drove classification performance, we computed topographic maps for each classifier set (early, mid, late). Weights resulting from backward decoding models are not interpretable as neural sources [99]. For that reason, we plotted topographic maps resulting from multiplying the data covariance matrix with the classifier weights, yielding activity patterns that are interpretable as neural sources underlying decoding results [95]. These covariance/class separability maps were then normalized across electrodes for each participant (mean activity over electrodes was thus zero).

In order to statistically evaluate classifiers' performance, we extracted diagonal and off-diagonal traces of GAT matrices. Average AUC values within specified clusters were analyzed using repeated measures ANOVA with factors decoding latency (early, late) and confidence (low, high). Specific hypotheses-driven comparisons between conditions were additionally evaluated using paired-sample $t$ tests on AUC values averaged within our TOI. To evaluate decoding off-diagonal, specifically, when the training was done using 150 to 200 ms classifiers that were then applied to the late 350 to 500 ms time-window, we compared average decoding in this time-window between 2 confidence levels (low versus high) using a paired-sample $t$ tests as well. In cases where a specific hypotheses-driven comparison did not indicate a significant result as indicated by an alpha of $p < 0.05$, we followed up that null-effect by a Bayesian equivalent of the same test in order to quantify the strength of evidence for the null hypothesis ($H_0$). By convention, Bayes factors from 1 to 3 were considered as anecdotal, 3 to 10 as substantial, and those above 10 as strong evidence in favor of $H_0$. Besides our hypothesis-driven analyses based on our predefined selection of time-windows, we additionally statistically compared each time point to chance (both for on- and off-diagonal decoding) by applying group-level permutation testing with cluster correction for multiple comparisons [100]. We employed two-tailed cluster-permutation testing with an alpha of $p < 0.05$ and cluster alpha $p < 0.05$ (N permutations = 1,000, as implemeted in ADAM toolbox). These latter analyses were mainly performed to evaluate whether our hypothesis-driven analyses may have missed

potentially relevant effects, and if so, we recommend future studies to replicate these effects to evaluate the robustness of them [98].

## Supporting information

**S1 Text. Additional results.** BF, Bayes factor; HC, high confidence; LC, low confidence. (DOCX)

**S1 Fig. Trial sequence of the sensory and decision localizer task.** (**A**) Each trial of the sensory localizer task started with a central red fixation dot for 1,200–1,400 ms, during which an infrequent contrast change of the dot could happen (20% of trials). Participants needed to press the spacebar as soon as they noticed the contrast change. In the same time interval, a house or a face image was briefly shown on the screen, which needed to be ignored. Images were either tilted to the right or to the left (task-irrelevant feature) at a 5˚ or 355˚ angle. Note that in the example trial, only a left-tilted house image is shown. (**B**) Each trial of the decision localizer task started with a central red fixation dot after which a forward mask was shown, followed by an image of a face or a house and a backward mask. Images were either tilted to the right or to the left (task-irrelevant feature) at a 5˚ or 355˚ angle. On every trial, participants reported whether they perceived a house or a face and indicated their confidence in this decision. (TIF)

**S2 Fig.** (**A**) GAT matrix for the decision localizer and regions of interest marked by inset black boxes (numbers 1–4). (**B**) Classifiers trained on the decision localizer applied to the main discrimination task (cross-task validation procedure). (**C**) GAT matrix for the sensory localizer. (**D**) Classifiers trained on the sensory localizer applied to the main discrimination task (cross-task validation procedure. (**E**) Classifiers trained on the sensory localizer applied to the decision localizer (cross-task validation procedure). (F) Classifiers trained on the decision localizer applied to the sensory localizer (cross-task validation procedure). All GAT matrices are based on occipital–parietal electrodes. The underlying data and scripts supporting this figure can be found on FigShare (https://doi.org/10.21942/uva.c.6265233.v1). (TIF)

**S3 Fig. Filtering effects on the timing of ERP components.** ERPs are time-locked to stimulus onset in the passive localizer with 3 different high-pass filtering settings. The region of interest plotted consists of the following electrodes: O1, O2, PO3, PO4, PO7, PO8. The underlying data and scripts supporting this figure can be found on FigShare (https://doi.org/10.21942/uva.c.6265233.v1). (TIF)

**S4 Fig. Face/house classification based on the decision localizer.** (**A**) On-diagonal time courses of correct and incorrect decisions, separated by confidence (LC, low confidence; HC, high confidence). Black boxes indicate the time-windows (early, mid, late) used for averaging the AUC scores. (**B**) Perceptual maintenance is derived by training a classifier on the time-window 150–200 ms and testing it across the entire time-window. In (A) and (B), colored horizontal lines indicate periods of significant decoding with respect to chance. Classification performance was evaluated at each time point using cluster-based permutation testing (two-tailed cluster-permutation, alpha $p < 0.05$, cluster alpha $p < 0.05$, N permutations = 1,000). (**C**) Bar plots showing average AUC values for the time-windows of interest highlighted in panels (A) and (B). The underlying data and scripts supporting this figure can be found on FigShare (https://doi.org/10.21942/uva.c.6265233.v1). (TIF)

**S5 Fig. Face/house classification based on the sensory localizer.** (**A**) On-diagonal time courses of correct and incorrect decisions, separated by confidence (LC, low confidence; HC, high confidence). Black boxes indicate the time-windows (early, mid, late) used for averaging the AUC scores. (**B**) Perceptual maintenance is derived by training a classifier on the time-window 150–200 ms and testing it across the entire time-window. In (A) and (B), colored horizontal lines indicate periods of significant decoding with respect to chance. Classification performance was evaluated at each time point using cluster-based permutation testing (two-tailed cluster-permutation, alpha $p < 0.05$, cluster alpha $p < 0.05$, N permutations = 1,000). (**C**) Bar plots showing average AUC values for the time-windows of interest highlighted in panels (A) and (B). The underlying data and scripts supporting this figure can be found on Fig Share (https://doi.org/10.21942/uva.c.6265233.v1).
(TIF)

## Author Contributions

**Conceptualization:** Josipa Alilović, Heleen A. Slagter, Simon van Gaal.

**Data curation:** Josipa Alilović, Simon van Gaal.

**Formal analysis:** Josipa Alilović.

**Funding acquisition:** Simon van Gaal.

**Investigation:** Josipa Alilović, Eline Lampers.

**Project administration:** Simon van Gaal.

**Supervision:** Heleen A. Slagter, Simon van Gaal.

**Visualization:** Josipa Alilović, Simon van Gaal.

**Writing – original draft:** Josipa Alilović, Simon van Gaal.

**Writing – review & editing:** Heleen A. Slagter.

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
