## [Editor Report · Decision Letter 0]

13 Jan 2022

Dear Simon, 

Thank you for submitting your manuscript entitled "The perceptual nature of illusory object recognition" for consideration as a Research Article by PLOS Biology.

Your manuscript has now been evaluated by the PLOS Biology editorial staff, as well as by an academic editor with relevant expertise, and I am writing to let you know that we would like to send your submission out for external peer review. Please accept my apologies for the delay in sending this decision to you.

Before we can send your manuscript to reviewers, we need you to complete your submission by providing the metadata that is required for full assessment. To this end, please login to Editorial Manager where you will find the paper in the 'Submissions Needing Revisions' folder on your homepage. Please click 'Revise Submission' from the Action Links and complete all additional questions in the submission questionnaire.

Once your full submission is complete, your paper will undergo a series of checks in preparation for peer review. Once your manuscript has passed the checks it will be sent out for review. To provide the metadata for your submission, please Login to Editorial Manager (https://www.editorialmanager.com/pbiology) within two working days, i.e. by Jan 17 2022 11:59PM.

If your manuscript has been previously reviewed at another journal, PLOS Biology is willing to work with those reviews in order to avoid re-starting the process. Submission of the previous reviews is entirely optional and our ability to use them effectively will depend on the willingness of the previous journal to confirm the content of the reports and share the reviewer identities. Please note that we reserve the right to invite additional reviewers if we consider that additional/independent reviewers are needed, although we aim to avoid this as far as possible. In our experience, working with previous reviews does save time. 

If you would like to send previous reviewer reports to us, please email me at ggasque@plos.org to let me know, including the name of the previous journal and the manuscript ID the study was given, as well as attaching a point-by-point response to reviewers that details how you have or plan to address the reviewers' concerns. 

Given the disruptions resulting from the ongoing COVID-19 pandemic, please expect some delays in the editorial process. We apologise in advance for any inconvenience caused and will do our best to minimize impact as far as possible.

Kind regards,

Gabriel

Gabriel Gasque

Senior Editor

PLOS Biology

ggasque@plos.org

---

## [Decision Letter · Decision Letter 1]

28 Feb 2022

Dear Dr van Gaal,

Thank you for submitting your manuscript entitled "The perceptual nature of illusory object recognition" for consideration as a Research Article at PLOS Biology. Thank you also for your patience as we completed our editorial process, and please accept my apologies for the delay in providing you with our decision. Your manuscript has been evaluated by the PLOS Biology editors, an Academic Editor with relevant expertise, and by three independent reviewers.

As you will see, the reviewers find the conclusions of your manuscript interesting, but they also raise several concerns that will need to be addressed in order to consider the manuscript further for publication. Both Reviewers 1 and 3 ask for several clarifications and offer some suggestions to improve the work, including the rewriting of some parts to have a better context in relation with previous literature. Reviewer 2 is more critical and raises some issues regarding the rationale of the study and the novelty of some of the findings, and also thinks that the link between confidence judgments and perceptual decisions should be better explored.

In light of the reviews (attached below), we will not be able to accept the current version of the manuscript, but we would welcome re-submission of a revised version that takes into account the reviewers' comments. We cannot make any decision about publication until we have seen the revised manuscript and your response to the reviewers' comments. Your revised manuscript is also likely to be sent for further evaluation by the reviewers.

We expect to receive your revised manuscript within 3 months. 

**IMPORTANT - SUBMITTING YOUR REVISION**

3. Resubmission Checklist

a) *PLOS Data Policy*

b) *Published Peer Review*

d) *Blurb*

Please also provide a blurb which (if accepted) will be included in our weekly and monthly Electronic Table of Contents, sent out to readers of PLOS Biology, and may be used to promote your article in social media. The blurb should be about 30-40 words long and is subject to editorial changes. It should, without exaggeration, entice people to read your manuscript. It should not be redundant with the title and should not contain acronyms or abbreviations. For examples, view our author guidelines: https://journals.plos.org/plosbiology/s/revising-your-manuscript#loc-blurb

Sincerely,

Ines

--

Ines Alvarez-Garcia, PhD

Senior Editor

PLOS Biology

on behalf of

Gabriel Gasque, PhD

Senior Editor

PLOS Biology

ggasque@plos.org

Reviewers' comments

Rev. 1: Guillaume Rousselet - note that this reviewer has signed his review

This is a very good article, presenting an ambitious and clever experiment, with thoughtful analyses. I have a lot of relatively minor comments and suggestions to try to improve the presentation, including illustrations and reporting.

## Paradigm

To check that the localisers did differ as intended, it would be useful to illustrate, maybe in Figure 1, cross-classification results: classification of localiser 1 data using the classifier trained with localiser 2 data, and vice versa.

## Illustrations

Figures 1-4: please replace the bar graphs with scatterplots and remove the little star system. P values are not measures of effect sizes or the importance of the results. If the goal is to emphasise effect sizes, it would be more informative to plot distributions of pairwise differences. This is essential to let readers assess how many participants show the effects, and the range of effect sizes. See suggestions here:

https://onlinelibrary.wiley.com/doi/full/10.1111/ejn.13400

Figures 2-4: please increase font size. Would be worth splitting the content of figures 2 and 3 across 2 figures to increase the size of all panels. Figure 4 could do with having 2 rows.

"As shown in Fig. 2E and 2G, diagonal decoding profiles were very different for correct and incorrect decisions" - this would be better reported by adding a new plot with correct and incorrect time-courses superimposed, and plotted separately for low and high confidence. Interactions could also be illustrated by plotting time-courses of differences.

I find this bit difficult to understand: "Most prominently, it can be observed that for correct trials, all processing stages show positive decoding, whereas in sharp contrast the two early stages for the incorrect trials showed negative decoding. This negative decoding pattern on incorrect decisions illustrates that early EEG data patterns look more similar to the presented stimulus category than to the reported stimulus category (note again that the decisions were used as the classifier labels)." By negative decoding, you mean below chance, not negative values? More importantly, could you provide a toy example, with simulated data from one participant, that would clearly demonstrate this dissociation and support your interpretation? Or is there a reference about this type of classifier's behaviour?

## Stat reporting.

The results section could be improved by going beyond outdated statements about the statistical significance of effects. Here are resources to more carefully report frequentist results, without falling into the trap of artificial dichotomies:

Language for communicating frequentist results about treatment effects

https://discourse.datamethods.org/t/language-for-communicating-frequentist-results-about-treatment-effects/934

Scientists rise up against statistical significance

https://www.nature.com/articles/d41586-019-00857-9

Inferential Statistics as Descriptive Statistics: There Is No Replication Crisis if We Don't Expect Replication

https://www.tandfonline.com/doi/full/10.1080/00031305.2018.1543137

Abandon Statistical Significance

https://www.tandfonline.com/doi/full/10.1080/00031305.2018.1527253

"Sensitivity (d') did not differ significantly from zero" - this suggests an equivalence test was performed. Otherwise, you can only conclude that you reserve judgement or that you did not get enough evidence. A simple statement would just state the results and point to a detailed graphical representation. There is no need to dichotomise all p values, unless you're trying to localise in space or time using brain imaging data - hard to see how to do that without some threshold.

In this sentence and possibly others, you seem to imply that the lack of evidence for an effect is evidence for its absence: "This was only the case when participants were highly confident in their incorrect decision (HC: t39=3.86, p<.001, d=0.61; LC: t39=1.41, p=.17, d=0.22, BF01=2.35; LC vs HC: t39=-3.06, p=.004, d=-0.48; Fig. 2H)". The statement should be revised by drawing attention to effect sizes and a supporting illustration. The absence of an effect cannot be derived from p values.

Same issue in lines 307-310 as well as other places, including supplementary information:

"Decoding strength was equal for correct and incorrect decisions (F1,39=0.06, p=.82, ηp2=0.001, BF01=5.7) and confidence did not modulate this decoding profile in any way (F1,39=0.54, p=.47, ηp2=0.014, BF01=4.77; interaction correctness x confidence: F1,39=0.18, p=.67, ηp2=0.005, BFexcl=4.22)."

Lines 270-272: can you provide values to qualify "much weaker"? For instance median difference + range?

275: "all other effects were identical" -> "were similar", as they cannot be identical.

808: please add a justification for your alpha.

## Other points

"To determine the nature of misreports, we present a novel way to dissociate sensory from post-sensory or decision-related processes during human perceptual decision-making using electroencephalography (EEG) in combination with multivariate pattern analyses (MVPA)." I'm struggling to see the novelty in the approach, so maybe you need to be more specific. These articles for instance, contain very similar ingredients:

https://jov.arvojournals.org/article.aspx?articleid=2121325

https://elifesciences.org/articles/68491

https://academic.oup.com/cercor/article/16/4/509/382221

https://www.jneurosci.org/content/26/35/8965

https://www.pnas.org/content/106/16/6539.short

Using EMG could also be a way to assess the motor stage:

https://hal.archives-ouvertes.fr/hal-01292369/document

In addition to refs 63-64, this is a good paper about mapping decoding results:

https://www.sciencedirect.com/science/article/pii/S1053811905003381

"N170 ERP component, specific to face processing" -- that component is also observed for letters, words, and biological motion for instance, so not specific to faces.

"In general, activity observed in sensory regions can thus be a mixture of both feedforward and feedback processes, especially at longer latencies (>~100 ms) (39)". Not sure if this has been confirmed and if it applies to non-moving stimuli, but feedback seems to affect the earliest V1 responses:

https://pubmed.ncbi.nlm.nih.gov/11152714/

See also this in humans:

https://pubmed.ncbi.nlm.nih.gov/11797091/

Both reinforcing your point, that indeed activity in sensory regions is a mixture of feedforward and feedback processes.

"were viewed with a distance of 90 cm from the monitor" - please indicate if a chin rest was used. If not, adjust text to something like "unconstrained viewing distance of approximately 90 cm".

663-5: "Furthermore, letters F and H, for faces and houses respectively, were shown on every trial in the right and left upper corner, or vice versa, depending on the block, as a as a reminder of the response mapping to the button press." For replicability, information is missing about the size, position and duration of these stimuli.

711-3: "EEG data was re-referenced to the average of the earlobes, high-pass filtered at 0.5 Hz and low-pass filtered at 40 Hz." Information is missing about the type of filters. Please also provide a figure illustrating filter distortions, ideally using a synthetic signal:

https://www.sciencedirect.com/science/article/pii/S0165027012002361

https://www.sciencedirect.com/science/article/pii/S0165027021000157

A gentle 0.5 Hz HP filter is unlikely to severely distort your temporal estimates, but it is worth checking and reminding readers about the dangers of HP filtering.

719: "All analyses were done using custom scripts..." - this would be a good place to mention that all the code and data are available online.

## Writing

The section starting line 101, "The time-course of category representations: Decision localizer", is difficult to read. It would help to refer to the little squares 1-4 in the figure. The squares are mentioned in a subsequent paragraph, but that description should be integrated with the earlier description. I would make figure 2A much larger and stand alone, to more carefully guide readers through it. Views of the diagonal, like in panels E and G are very useful, and would help support the text description and the understanding of panel A.

"this neural representation "flipped" later in time" - avoid the quotation marks of mystery, be specific instead. Same for "global ignition" and other expressions, unless they are quotes.

This sentence is hard to read:

"Here we investigated this issue and examined at what level of analysis visual stimuli that were misreported (e.g., face presented, house reported) were actually misrepresented, at the neural level, aiming to dissociate sensory from decision-related explanations."

Suggested revision:

"Here we investigated this issue and examined if misreported visual stimuli (e.g., face presented, house reported) were misrepresented at the sensory or the decision stage."

"As can be seen in Fig. 1, the orientation of the presented images was either left tilting or right tilting, with 50/50 likelihood." In Figure 1 all the images are tilted to the left.

Lines 108-109: "In contrast, generalization across time analyses can" - delete "analyses"? That whole sentence needs to be rephrased to remove the repetition "generalization across time [...] observed when a classifier trained on certain time points generalizes to other data points earlier or later in time".

Line 112: "Category-specific neural representations could be robustly decoded" -- robust to what? I would delete "robust".

Figure 2 caption: "On the right covariance/class separability" - is a comma missing after "right"?

Line 245: "despite never been objectively presented and never been" -> "despite never being presented and never being..." Not sure what an "objective presentation" means. Would be clearer to state "presented on the screen".

803: "covariance/class separability map maps were" - extra "map"?

Signed review: Guillaume Rousselet

Rev. 2:

The manuscript is a report of an EEG experiment where participants were asked to discriminate images of faces and houses near threshold. Multivariate analyses are performed and reveal that early sensory processing is consistent with the physical stimulus whereas later processing is consistent with the percept. The experiment is cleanly designed and analyses, and the manuscript is well-written. But I am not convinced that the results are sufficiently ground-breaking to appear in a highly selective journal like PLOS Biology.

My first concern is that the motivation of the study is overlooking decades of research that have addressed the same question as the authors. This is rather surprising given that the senior authors have themselves largely contributed to this research. Just to give a few examples, Linares et al. (2019) have presented a method to decouple sensory from decisional choice biases in perceptual decision making, Kunimoto et al. (2001) have presented behavioral results on the link between near-threshold perceptual responses and confidence judgments, and Fleming & Dolan (2012) have presented the neural bases for these behavioral results.

My second concern is that the novelty of the results is exaggerated. The authors seem surprised that early sensory processing is consistent with the physical stimulus whereas later processing is consistent with the percept. How could it be otherwise? Again, the neural basis of this hierarchical processing of ambiguous information has been established decades ago (see e.g. Logothetis & Schall, 1989).

My third concern is that the link between confidence judgments and perceptual decisions is only superficially explored. For instance, the four-point rating scale is analyzed as just high versus low confidence, disregarding that the lowest confidence level was linked to chance perceptual performance (Figure 1B). Some analyses commonly found in the metacognition literature (for instance using meta-d'; Maniscalco & Lau, 2012) would have been welcomed. There are also interesting EEG-based analyses worth considering on this topic (e.g. Charles et al., 2013; Balsdon et al., 2021).

Minor comments:

- Figures 2E, F, G, & H: would it be possible to use the same scale on the y-axis to better compare classification performance between correct and incorrect decisions?

- line 213: replace "Figure 3E" by "Figure 2E"

References

Balsdon, T., Mamassian, P. & Wyart, V. (2021). Separable neural signatures of confidence during perceptual decisions. ELife, 10, e68491. https://doi.org/10.7554/elife.68491

Charles, L., Van Opstal, F., Marti, S., & Dehaene, S. (2013). Distinct brain mechanisms for conscious versus subliminal error detection. NeuroImage, 73, 80-94. https://doi.org/10.1016/j.neuroimage.2013.01.054

Fleming, S. M. & Dolan, R. J. (2012). The neural basis of metacognitive ability. Philosophical Transactions of the Royal Society of London Series B, Biological Sciences, 367(1594), 1338-1349. https://doi.org/10.1098/rstb.2011.0417

Kunimoto, C., Miller, J., & Pashler, H. (2001). Confidence and accuracy of near-threshold discrimination responses. Consciousness and cognition, 10(3), 294-340. https://doi.org/10.1006/ccog.2000.0494

Linares, D., Aguilar-Lleyda, D. & López-Moliner, J. (2019). Decoupling sensory from decisional choice biases in perceptual decision making. ELife, 8, e43994. https://doi.org/10.7554/elife.43994

Logothetis, N. K., & Schall, J. D. (1989). Neuronal correlates of subjective visual perception. Science, 245(4919), 761-763. https://doi.org/10.1126/science.2772635

Maniscalco, B. & Lau, H. C. (2012). A signal detection theoretic approach for estimating metacognitive sensitivity from confidence ratings. Consciousness and Cognition, 21(1), 422-430. https://doi.org/10.1016/j.concog.2011.09.021

Rev. 3: Paul Sajda - note that this reviewer has signed his review

In this paper, Aliliovic et al investigate how misinterpretations of sensory stimuli are reflected in EEG activity using an MVPA approach that separates sensory and decision related processes in the neural activity. They find that when subjects are confident in a decision which is an "error" (i.e. their choice does not match the stimulus category) sensory stages of stimulus processing represent the category and then later flip to a representation of the choice category. Note the reason this is important is that it is only seen for the high confidence, erroneous decisions. This work adds to the literature on how sensory evidence is transformed into a decision and how confidence may arbitrate bottom up and top down process.

I very much enjoyed the paper and I think it adds to the literature. I have some comments below that I think might help clarify the presentation a bit, particularly within the context of prior work. To be completely transparent most of the prior work I will reference below I am a co-author, and though I am asking questions related to this prior work, by no means do I require/expect the authors to reference all this work unless they believe it relevant. However there are a number of interesting findings they report that map nicely to some of my prior work that I think would be nice to reconcile/consider. I sign my name at the bottom of this review so that the authors know who I am.

1. Interaction between sensory and decision making processes explain differences between stimulus category and choice.

The interaction and "flipping" of neural representation for high confidence error trials is a very interesting finding showing the interaction between confidence, choice and sensory input. in [1] our group used a similar task (discriminating faces from houses or cars) and simultaneously recorded EEG and fMRI and showed that differences in interactions between "early" and "late" networks could explain decision bias. Note that no reports of confidence were used in our study, however bias was the measure that was representative of the mismatch between stimulus and choice category. I think that prior work and this new paper presented by the authors together show that there is clearly a bottom up and top down interaction occurring when "perpeptual illusions" or "decision bias" is in play -- i.e. when the sensory category does not match the choice category and individuals are confident in their choice. It would be interesting to know the authors thoughts with respect to this prior work. Also, as an aside, did you observe any face bias for the decision errors?

2. Separating sensory and decision processes.

This of course is key to the work presented here. I would recommend adding a little more detail about how the MVPA was trained/optimized and also on interpretation (see below). Also some other work that might be worth considering (though once again just a suggestion) which supports these types of approaches to disassociate different processing stages in perceptual decision making.

[2] used MVPA classifiers on EEG for a face vs car task to separate sensory and evidence related signals. Together with [3] this work showed that early discriminating components are likely related to an iconic memory or "perceptual maintenance stage" (see your manuscript, line 133), and that in fact when comparing the single trial variability of these components using a DDM [4] the components clearly dissociate into a sensory evidence variable and decision/choice evidence variable. [5] uses a MVPA technique to show that multiple decision related signals are expressed across a perceptual decision and that they can be tracked via analysis of the MVPA output across time. This prior work just reinforces the approach the authors take and shows that one can use MVPA to dissociate neural correlate of perceptual and cognitive processes.

3. MVPA classifier

I think it would be helpful to know a little more about the classifier. For example, is it a logistic regression classifier, LDA, etc. Was the optimization regularized? In the supplementary material it would be good to show mathematical expressions for the classifier as well as the equation for what is being optimize and if this is a convex optimization or is regularized in some way. In Fig2 and 3, panel A, plotting significant channels is interesting but it might be better to plot forward models which take into account the multivariate interactions between channels and would also be more interpretable.

4. Minor comments

Figure 1B should be clarified. Is significance between confidence levels relative to d' =0 or is this significance relative to one another? It is clear that for confidence = 1, that this is respect to d' = 0, but not clear for the other confidence levels.

The sensory localizer also seems to be an oddball task (in frequent (1/5) contrast change). Perhaps mention how this will add an orienting response to the sensory localizer component. I don't think it is problematic per se, but it adds another process into what the sensory localizer represents.

References cited above

[1] Tu, T., Schneck, N., Muraskin, J., & Sajda, P. (2017). Network configurations in the human brain reflect choice bias during rapid face processing. Journal of Neuroscience, 37(50), 12226-12237.

[2] Philiastides, M. G., Ratcliff, R. & Sajda, P. Neural representation of task difficulty and decision making during perceptual categorization: a timing diagram. The Journal of neuroscience : the official journal of the Society for Neuroscience 26, 8965-8975 (2006).

[3] Philiastides, M. G. & Sajda, P. EEG-informed fMRI reveals spatiotemporal characteristics of perceptual decision making. The Journal of neuroscience : the official journal of the Society for Neuroscience 27, 13082-13091 (2007).

[4] Ratcliff, R., Philiastides, M. G. & Sajda, P. Quality of evidence for perceptual decision making is indexed by trial-to-trial variability of the EEG. Proceedings of the National Academy of Sciences of the United States of America 106, 6539-6544 (2009).

[5] Philiastides, M. G., Heekeren, H. R. & Sajda, P. Human Scalp Potentials Reflect a Mixture of Decision-Related Signals during Perceptual Choices. The Journal of neuroscience : the official journal of the Society for Neuroscience 34, 16877-16889 (2014).

signed Paul Sajda

---

## [Decision Letter · Decision Letter 2]

7 Dec 2022

Dear Simon,

Thank you for your patience while we considered your revised manuscript "The perceptual nature of illusory object recognition" for publication as a Research Article at PLOS Biology. This revised version of your manuscript has been evaluated by the PLOS Biology editors, the Academic Editor, and the original reviewers.

As the reviewers are now satisfied with the revisions, we are likely to accept this manuscript for publication. At this point, we simply ask that you address a few editorial requests and the required data and other policy-related requests detailed at the bottom of this email.

Editorial Requests:

***Article title - Please consider a modified title that will more completely convey the interesting findings in this work. We'd suggest something like: 

Illusory object recognition reflects either sensory perceptual or cognitive origins depending on decision confidence

**Blurb – 

Please also provide a blurb which, if the paper is accepted, will be included in our weekly and monthly Electronic Table of Contents (eTOCs), sent out to readers of PLOS Biology. This blurb may also be used to promote your article on social media. The blurb should be about 30-40 words long and is subject to editorial changes. It should, without exaggeration, entice people to read your manuscript, should not be redundant with the title and should not contain acronyms or abbreviations. For examples, view our author guidelines: https://journals.plos.org/plosbiology/s/revising-your-manuscript#loc-blurb

As you address these items, please also take this last chance to review your reference list to ensure that it is complete and correct. If you have cited papers that have been retracted, please include the rationale for doing so in the manuscript text, or remove these references and replace them with relevant current references. Any changes to the reference list should be mentioned in the cover letter that accompanies your revised manuscript.

We expect to receive your revised manuscript within two weeks. 

*Published Peer Review History*

*Press*

Sincerely,

Kris

Kris Dickson, Ph.D., (she/her)

Neurosciences Senior Editor/Section Manager,

kdickson@plos.org,

PLOS Biology

DATA POLICY:

As you are aware, the PLOS Data Policy requires that all data be made available without restriction: http://journals.plos.org/plosbiology/s/data-availability. For more information, please also see this editorial: http://dx.doi.org/10.1371/journal.pbio.1001797

***We note that the DOI link provided on Figshare (https://doi.org/10.21942/uva.c.6265233) is not active. This will need to be updated to allow our readership full access to the underlying summary data.

1) Deposition in a publicly available repository (such as Figshare or Zenodo). Please also provide the accession code or a reviewer link so that we may view your data before publication. 

2) Supplementary files (e.g., excel). Please ensure that all data files are uploaded as 'Supporting Information'.

In either case, please make sure the files are invariably referred to (in the manuscript, figure legends, and the Description field when uploading your files) using the following format verbatim: S1 Data, S2 Data, etc. Multiple panels of a single or even several figures can be included as multiple sheets in one excel file that is saved using exactly the following convention: S1_Data.xlsx (using an underscore).

***Regardless of the method selected, please ensure that you provide the individual numerical values that underlie the summary data displayed in the following figure panels as they are essential for readers to assess your analysis and to reproduce it:

Fig 1B; Fig 2A,B; Fig 3A-E; Fig 4A-E; Fig 5A-C; SuppFig 2A-F; SuppFig 3; SuppFig 4A-C; SuppFig 5A-C

***Please also ensure that figure legends in your manuscript and in the supplemental file are updated to include information on WHERE THE UNDERLYING DATA CAN BE FOUND (e.g. “The underlying data supporting Fig X, panel Y can be found in file Z.”). Please also ensure that your supplemental data file/s has a legend.

Please ensure that your Data Statement in the submission system accurately describes where your data can be found, with the DOI link being updated to an active file link.

DATA NOT SHOWN?

- Please note that per journal policy, we do not allow the mention of "data not shown", "personal communication", "manuscript in preparation" or other references to data that is not publicly available or contained within this manuscript. Please double check your submission and either remove mention of any such data or provide figures presenting the results and the data underlying such figure(s).

Reviewer remarks:

Reviewer's Responses to Questions

Do you want your identity to be public for this peer review?

Reviewer #1: Yes: Guillaume Rousselet

Reviewer #2: No

Reviewer #3: Yes: Paul Sajda

Reviewer #1: The authors have tackled all my comments. I've also read the reply to the other reviewers and I think the new version addresses all their comments as well. Emphasis on novelty is not something I consider important, but the authors have now clearly articulated how their work builds upon previous studies on a similar topic. The revision made an already strong article even stronger: the writing is sharper, the reporting and figures more detailed and informative, and the interpretations more nuanced. I would change the wording of a few things, especially in the results, but that's down to personal preferences, so no need to make any more changes at this stage. The authors should be really proud of what they have achieved here. 

Reviewer #2: I would like the authors for taking all the reviewers' comments seriously, including mine, and for writing a detailed response letter. The changes made to the manuscript in the text and the figures are very much welcomed. The analogy to the no-report paradigm definitely helped me clarify a few points. 

On the use of confidence reports, the authors are stating that "confidence was not used as a dependent measure per se" (response letter), but in the abstract, they admit "This work demonstrates that decision confidence arbitrates between perceptual decision errors, which reflect true illusions of perception, and cognitive decision errors, which do not". So it seems that confidence was after all an important dependent variable in their design. However, I agree with the authors that an in-depth analysis of this variable would have made this manuscript much longer and more cumbersome.

In summary, I think the results are clean, the analyses are trust-worthy, and thus this manuscript deserves to be nicely published.

Reviewer #3: The reviewers have addressed all my comments and concerns and I recommend the manuscript be accepted. 

Signed: Paul Sajda

---

## [Editor Report · Decision Letter 3]

20 Jan 2023

Dear Dr van Gaal,

Thank you for the submission of your revised Research Article "Illusory object recognition is either perceptual or cognitive in origin depending on decision confidence" for publication in PLOS Biology. On behalf of my colleagues and the Academic Editor, Chris Pack, I am pleased to say that we can in principle accept your manuscript for publication, provided you address any remaining formatting and reporting issues. These will be detailed in an email you should receive within 2-3 business days from our colleagues in the journal operations team; no action is required from you until then. Please note that we will not be able to formally accept your manuscript and schedule it for publication until you have completed any requested changes.

PRESS

We frequently collaborate with press offices. If your institution or institutions have a press office, please notify them about your upcoming paper at this point, to enable them to help maximize its impact. If the press office is planning to promote your findings, we would be grateful if they could coordinate with biologypress@plos.org. If you have previously opted in to the early version process, we ask that you notify us immediately of any press plans so that we may opt out on your behalf.

Sincerely, 

Kris

Kris Dickson, Ph.D., (she/her)

Neurosciences Senior Editor/Section Manager

PLOS Biology

kdickson@plos.org